# The Impact of Chronic Kidney Disease on Nutritional Status and Its Possible Relation with Oral Diseases

**DOI:** 10.3390/nu14102002

**Published:** 2022-05-10

**Authors:** Micaela Costacurta, Michele Basilicata, Giulia Marrone, Manuela Di Lauro, Vincenzo Campolattano, Patrizio Bollero, Raffaella Docimo, Nicola Di Daniele, Annalisa Noce

**Affiliations:** 1Pediatric Dentistry, Department of Surgical Sciences, University of Rome Tor Vergata, 00133 Rome, Italy; micaela.costacurta@uniroma2.it (M.C.); raffaelladocimo@tiscali.it (R.D.); 2UOSD Special Care Dentistry, Department of Experimental Medicine and Surgery, University of Rome Tor Vergata, 00100 Rome, Italy; michele.basilicata@ptvonline.it; 3UOC of Internal Medicine-Center of Hypertension and Nephrology Unit, Department of Systems Medicine, University of Rome Tor Vergata, 00133 Rome, Italy; giulia.marrone@uniroma2.it (G.M.); dilauromanuela@gmail.com (M.D.L.); didaniele@med.uniroma2.it (N.D.D.); 4UOSD Special Care Dentistry, Department of Systems Medicine, University of Rome Tor Vergata, 00100 Rome, Italy; vincenzo.campolattano@students.uniroma2.eu (V.C.); patrizio.bollero@ptvonline.it (P.B.)

**Keywords:** chronic kidney disease, periodontal diseases, hypovitaminosis, protein-energy wasting, metabolic acidosis, low-grade inflammation, high-protein diet

## Abstract

Several studies have demonstrated a strong relation between periodontal diseases and chronic kidney disease (CKD). The main mechanisms at the base of this link are malnutrition, vitamin dysregulation, especially of B-group vitamins and of C and D vitamins, oxidative stress, metabolic acidosis and low-grade inflammation. In particular, in hemodialysis (HD) adult patients, an impairment of nutritional status has been observed, induced not only by the HD procedures themselves, but also due to numerous CKD-related comorbidities. The alteration of nutritional assessment induces systemic manifestations that have repercussions on oral health, like oral microbiota dysbiosis, slow healing of wounds related to hypovitaminosis C, and an alteration of the supporting bone structures of the oral cavity related to metabolic acidosis and vitamin D deficiency. Low-grade inflammation has been observed to characterize periodontal diseases locally and, in a systemic manner, CKD contributes to the amplification of the pathological process, bidirectionally. Therefore, CKD and oral disease patients should be managed by a multidisciplinary professional team that can evaluate the possible co-presence of these two pathological conditions, that negatively influence each other, and set up therapeutic strategies to treat them. Once these patients have been identified, they should be included in a follow-up program, characterized by periodic checks in order to manage these pathological conditions.

## 1. Introduction

Periodontitis is a chronic multifactorial inflammatory disease associated with dysbiosis of the oral flora, characterized by the progressive breakdown of the dental support apparatus, with loss of clinical attachment, loss of alveolar bone, presence of periodontal pockets and gingival bleeding [1]. Periodontitis is a serious public health issue, as it can cause not only local symptoms, but it can also have a negative impact on the individual general health, contributing to the development, and to the worsening, of chronic non-communicable degenerative diseases, like chronic kidney disease (CKD) [2].

The prevalence of periodontitis in CKD patients differs considerably in literature, as indicated in a review by Serni et al., which reported a wide range of prevalence, from 34.35 to 93.65%. CKD was associated with a high risk of periodontitis. In particular, CKD stages IV-V patients present a higher risk of developing periodontitis compared to healthy subjects. At the oral cavity level, the CKD patients may have higher mean probing pocket depth (PPD), and clinical attachment level (CAL) values with an average difference in PPD and CAL of 0.25 mm, (*p* = 0.02) and 0.41 mm (*p* < 0.0001), respectively, compared to healthy subjects [3].

The progression of untreated and non-intercepted periodontitis leads to tooth loss in end-stage renal disease (ESRD) [4,5], resulting in alterations in chewing, phonetics and aesthetics, loss of self-esteem, impaired quality of life and relevant public health costs. A cross-sectional study has explored the associations between periodontitis and health-related quality of life (HRQoL) among hemodialysis (HD) patients, dividing the participants into three groups (severe, moderate, and no/mild periodontitis). The results showed that HRQoL decreased among HD patients with severe periodontitis [6].

It has been stated that there is a possible bidirectional depend/mutual relationship between CKD and periodontal diseases (PD). Fisher and Taylor demonstrated that PD is a risk factor for the onset of CKD in a study on 11,955 adult people in the United States [7] and this result was confirmed by a systematic review. Moreover, the treatment of PD seems to impact positively on the glomerular filtration rate (GFR) [8]. Ioannidou and Swede observed a direct correlation between the severity of PD and the stage of CKD and they demonstrated that CKD patients were 30–60% more likely to develop moderate PD [9].

Later, another study, also conducted by Ioannidou et al., showed that Mexican Americans with a lower renal function had a double chance of developing PD than individuals with normal renal function [10]. In the meantime, Iwasaki et al. proved the link between PD and lower renal function in old Japanese people [11]. PD is associated with the risk of atherosclerotic vascular disease in HD [12]. Moreover, mortality in HD patients is related to PD [13,14], to poor oral hygiene and to dental caries [15]. In a recent prospective cohort study, with a 14-year follow up, Ricardo et al. highlighted that CKD patients with PD were 35% more likely to die than those without PD [16].

Among the factors related to the onset of CKD in PD patients and vice versa, malnutrition, vitamin dysregulation, especially of B-group vitamins and of C and D vitamins, oxidative stress (OS), metabolic acidosis and low-grade inflammation (LGI) seem to have key functions (Figure 1) [17,18,19,20,21,22,23]. These factors may be related to the stage of CKD, and they are particularly exacerbated in ESRD, namely the final stage of CKD, which often requires renal replacement therapy, such as dialysis or kidney transplantation [24].

The aims of this review are:to assess alterations of nutritional status and vitamin dysregulation in HD patients regarding oral diseases and *vice versa*;to evaluate the role of metabolic acidosis and low-grade inflammation (LGI) in CKD patients on oral diseases and *vice versa*.

## 2. Search Methods

All the articles selected for this review were acquired using searches strategies on PubMed and Scopus databases up until March 2022. The outcome of interest was the impact of impaired nutritional assessment in HD patients on oral health. In the search strategy, the keywords consisted of hemodialysis, protein energy wasting (PEW) syndrome, hypovitaminosis, metabolic acidosis, low-grade inflammation, periodontal diseases, oral health, oral diseases and chronic periodontitis (hemodialysis OR chronic kidney disease OR end-stage renal disease) AND (PEW syndrome OR hypovitaminosis OR metabolic acidosis OR low-grade inflammation) AND (periodontal diseases OR oral health OR chronic periodontitis OR oral diseases). All articles were manually retrieved by the authors and only English language papers and journal were selected (Figure 2).

## 3. Alterations of Nutritional Status in Hemodialysis Patients

Hemodialysis (HD) treatment further alters the patient’s nutritional status, already compromised by the end-stage renal disease (ESRD) condition. In particular, it is estimated that almost one third of HD patients present a moderate degree of malnutrition and that almost 10% present severe malnutrition [25]. HD patients, in fact, frequently exhibit a reduction in body weight, mainly linked to muscle mass loss and lower levels of serum albumin, transferrin, pre-albumin and other biochemical parameters [26]. These conditions are also exacerbated by the HD patient’s clinical and demographic features, such as advanced age and the presence of cardiovascular (CV) and gastrointestinal diseases, such as uremic gastritis. All these conditions induce a reduction of quality of life and an increased risk of mortality for all causes [27].

Among the causes of malnutrition, it is possible to observe:(i)inadequate caloric and protein intake (protein-energy wasting, PEW) and typical CKD-related comorbidities (like metabolic acidosis, uremic gastritis, anorexia, depression, etc.);(ii)chronic systemic inflammatory state and complications typical of HD treatment (such as hypercatabolic state, loss of protein and amino acids during the dialytic session, etc.) [28].

PEW syndrome can be secondary to factors related to uremia, including nausea, gastroesophageal reflux, decreased appetite and negative nitrogen balance, and to the patient’s psychosocial conditions (i.e., non-acceptance of the chronic disease, depression, etc.) and unhealthy eating habits [29]. In fact, patients may tend to maintain pre-dialysis eating habits, characterized by a low-protein intake, which tends to exacerbate the malnutrition state [30]. When CKD reaches the ESRD, the risk of developing metabolic and nutritional derangements is very high. These conditions, if not counterbalanced, give rise to this syndrome. Firstly, in 1983, the World Health Organization defined, the term “wasting” as an unintentional body weight loss >10% in the absence of opportunistic infection, cancer, or chronic diarrhoea [31]. The “wasting” condition is common in the CKD population. The diagnostic criteria proposed by the International Society of Renal Nutrition and Metabolism (ISRNM) for PEW detection are based on the presence of at least three of the following listed categories: (i) laboratory parameters such as serum albumin < 3.8 g/L, (ii) body mass index lower than 23 kg/m^2^ or unintentional weight loss over time, of at least 5% over three months or 10% over six months, (iii) reduced muscle mass of 5% over three months or of 10% over six months and (iv) protein intake <0.8 g/kg of ideal body weight (i.b.w.) per day for at least two months [32].

Kopple et al. [31] reported that the prevalence of PEW syndrome is indicated in between 28–80% of HD patients. Several studies demonstrated that PEW is the major determinant of poor clinical outcomes, impacting on risk of hospitalization, frailty and death for CV disease in these patients [33]. Multiple factors are linked to PEW syndrome and among these are: (i) high levels of uremic toxins [34], (ii) catabolic status associated to HD [33], (iii) abnormalities in mineral metabolism, such as secondary hyperparathyroidism, and acid-base unbalance like metabolic acidosis [17], (iv) comorbidities, such as CV diseases, gastrointestinal diseases and depression [35], (v) inflammatory status [36] and (vi) poor nutritional intake [37]. Furthermore, PEW syndrome can be induced and exacerbated by metabolic acidosis. The latter is a clinical condition characterized by an acid-base imbalance, in which it is observed an accumulation of hydrogen ions and a reduction of serum bicarbonate concentration (<22 mmol/L) with consequent blood acidification [17]. Metabolic acidosis can induce muscle mass loss through proteolysis stimulation at the skeletal muscles level. In particular, this condition seems to be able to stimulate the ATP-ubiquitin-proteasome pathway and, at the same time, inhibit, even if minimally, protein synthesis [38,39]. Another clinical manifestation present in CKD patients is represented by the co-occurrence of malnutrition, inflammation and atherosclerosis (MIA), which characterizes the clinical condition called MIA syndrome, more frequently found in ESRD patients [40]. In fact, stage V CKD patients are subjected to a series of food restrictions in order to avoid potassium, phosphorus and sodium accumulation [41]. However, patients are not always compliant with the nutritional restraints and they often refuse the nutritional approaches, or, conversely, they strictly adhere to the diet, eliminating most foods and consequently introducing a low caloric intake. In this context, it is common the onset of malnutrition, and a nutritional evaluation should be conducted by an experienced renal nutritionist [42,43]. The pathological mechanisms that cause MIA syndrome are also attributable to reduced kidney ability to maintain blood acid-base homeostasis, as mentioned above [44]. The acid uremic milieu triggers the production of tumor necrosis factor-α (TNF-α) by macrophages. In an in vitro study, an increased release of TNF-α was demonstrated after incubation of peritoneal macrophages in an acidic cell culture medium, hypothesizing a possible relationship between LGI and metabolic acidosis [45]. This condition contributes to the chronic inflammatory status, typical of CKD, inducing a vicious cycle [46,47]. Malnutrition and chronic inflammatory status have been strongly associated with atherosclerosis. Vascular calcifications are common in HD patients and they determine a higher risk of CV morbidity and mortality [48]. The typical manifestations of calcium-phosphorus unbalance are not the formation of plaques, but the artery calcifications and the transformation of smooth muscle cells of the arterial wall into osteoblast-like cells [49,50]. A study conducted by Choi et al. concluded that the combination of malnutrition and inflammation significantly impacted on the onset of abdominal aortic calcification. Moreover, they suggested the need of studying the combined effects of anti-inflammatory and nutritional interventions on vascular calcifications prevention [51].

Another possible cause of malnutrition in HD patients is related to the dialysis treatment itself. In fact, during the dialytic session, a condition of protein hyper-catabolism occurs, due to the intradialytic loss of amino acids and the production of pro-inflammatory cytokines [52]. In particular, it is estimated that during each HD session up to about 10–12 g of amino acids are lost in the dialysate and this loss is increased with the use of high-flux membranes [53,54,55]. Moreover, dialysis treatment induces an increase in inflammatory risk factors, including local infections in the arteriovenous fistula or in the central venous catheter for HD. Other inflammatory risk factors are related to membrane biocompatibility and to dialysate quality [56]. Loss of amino acids in the dialysate, and increase in circulating inflammatory cytokines, are responsible, furthermore, for muscle mass impairment. Moreover, pro-inflammatory cytokines act at the level of the central nervous system by altering the appetite/satiety balance, inducing anorexia [57].

## 4. Nutritional Therapy in Hemodialysis Patients

In HD patients, the prescription of a personalized nutritional therapy plays a pivotal role in clinical management. The common feature of nutritional therapies is the control of micronutrient intake (such as potassium, phosphorus and sodium) and of liquids in general. Another aspect to take into consideration in the nutritional plans concerns protein intake, which must be >1.1–1.2 g of protein/kg i.b.w/day, as well as caloric intake (~30–35 kcal/kg of i.b.w./day). The latter, if not adequately balanced, can give rise to PEW syndrome [58]. For this purpose, nutritional management in HD patients decisively impacts on their prognosis and quality of life. In fact, malnutrition, present in HD patients, negatively affects public health care costs, causing patient falls, bone fractures and frequent and long-term hospitalizations [59].

Among nutritional restrictions for HD patients, a low-potassium intake (lower than 2.7–3.1 g/day) [60] permits a decrease in the risk of malignant arrythmias and CV mortality [61] due to potassium overload. Potassium is a micronutrient commonly present in plant-based foods, such as fruit and vegetables, and its uncontrolled intake could cause CV impairments. For this purpose, it is important to avoid foods rich in potassium and to use soaking and boiling to obtain demineralization of this micronutrient. Additional attention should be paid to hidden potassium sources, such as foods with additives and salt substitutes (like low-sodium salts) [62]. With renal function loss, phosphorus tends to accumulate in the bloodstream and, furthermore, impairments in calcium, vitamin D, parathormone and fibroblast growth factor-23 (FGF-23) concentration levels can be observed. These metabolic alterations trigger the CKD-mineral bone disorder (CKD-MBD) [63] that dramatically impacts on the survival and life quality of ESRD patients. In fact, when the calcium-phosphorus product is higher than 55 mg^2^/dL^2^ [64,65], it gives rise to “calciprotein particles” formed by colloidal nanoparticles of calcium phosphate [66]. These compounds precipitate into small and large caliber vessels inducing CV calcifications. As previously said, high serum phosphorus levels trigger the transformation of smooth muscle cells of the arterial wall into osteoblastic-like cells, which increase vascular rigidity and predispose to heart failure [67]. Uremic patients are often characterized by high phosphorus levels, due to exceeding net intestinal absorption, low renal excretion and poor removal during dialysis procedures [68]. For these reasons, a nutritional approach aimed at controlling phosphorus intake is crucial for uremic patients’ clinical management. Phosphorus can be ingested through inorganic or organic sources. Phosphorus is mostly found in food additives (such as phosphoric acid, phosphates, etc.) commonly used in processed and industrial foods to increase their color and flavor over time. Regarding organic phosphorus, it should be ingested from animal or planted-based foods, such as meat, fish, poultry, dairy products and from fruit, vegetables and grains, as well. Among the three different phosphorus sources, the inorganic ones are mostly absorbable (up to 90%), followed by animal and plant-based foods (up to 60% vs. 40%) [69]. Even for organic phosphorus, soaking and boiling could help reduce its content [70]. Physiologically, despite observed variations in daily water and sodium intake, the volemia tends to remain constant, thanks to a series of mechanisms that act by regulating thirst and urinary excretion of water and sodium. If the renal function is impaired, an altered capacity for sodium excretion by the kidney was observed. In CKD, the inability to excrete sodium represents the most important cause of arterial hypertension (AH) and of increased volemia. This condition can lead to renal disease progression [71]. To maintain homeostasis it is advisable, for ESRD patients, to assume a maximum of 2–3 g of sodium *per* day [72]. Moreover, in ESRD patients, the dialysis procedure is the only chance to remove excess dietary salt intake. If there is a mismatch between sodium removal and its intake, fluid overload, AH, and left ventricular hypertrophy, causing pulmonary edema, could be observed [73]. In HD patients, fluids should be reduced to prevent circulatory overload and pulmonary edema [74]. In fact, ESRD patients under HD treatment should drink 500–700 mL of water *per* day, if they are anuric. If they have residual diuresis, they should drink 500–700 mL of water *per* day plus the residual urine volume [75].

One of the most important goals for HD patients is adequate protein and caloric intake. Current nutritional guidelines suggest a daily caloric intake that should be almost 30–35 kcal/kg of i.b.w. and a daily protein intake higher than 1.1–1.2 g/kg of i.b.w., to prevent defective nutrition due to the HD procedure itself [76]. HD patients might present two different kinds of malnutrition, respectively: (i) poor nutritional intake or (ii) impaired body composition, with a reduction of the systemic proteins pool as a consequence of inflammatory status [77]. To avoid loss of muscle mass and body weight, it is important that a HD patient regularly undergoes nutritional assessment in order to prevent and to counteract the onset of malnutrition [78].

## 5. Alterations of Vitamins Levels in CKD Patients and Their Influence on Oral Diseases

During dialysis treatment, there is not only a loss of amino acids, but also of important micronutrients, such as vitamins and minerals. The main vitamin deficiencies often found in HD patients concern vitamin C, folate and vitamin D [79]. In addition to the loss of vitamins during dialysis treatment, there are several mechanisms that can induce a vitamin deficiency (Figure 3). One of the main causes is certainly attributable to dietary-nutritional restrictions (primarily concerning fresh fruit and vegetables, rich in vitamins and minerals) and to incorrect nutritional intake (mainly due to anorexia related to uremic gastropathy) [80]. On the other hand, other mechanisms are attributable to impaired vitamin metabolism (as occurs for folate), to their incorrect synthesis or activation (as observed for vitamin D) or to their reduced intestinal absorption (as occurs for vitamin C and other B-group vitamins) [81,82,83].

### 5.1. The Role of Vitamin C

Physiologically, vitamin C is mainly absorbed in the small intestine by an active transport mechanism (Na+-dependent) and by passive diffusion through the concentration gradient. The first mechanism takes place in the case of low vitamin C levels, whereas, the second occurs in the case of upper levels. Successively, it is carried to different systems and apparatus through the blood and partially stored in the liver and adrenal glands. In healthy subjects, the disproportionate amount of vitamin C, in relation to needs, is filtered by kidneys and excreted in the urine, both unmodified and in the form of metabolites [84,85].

The absence of an enzyme called L-gluconolactone oxidase means humans and monkeys are unable to produce ascorbic acid [86]. Due to the fact that vitamin C is not produced in our bodies and it cannot be stored, a correct daily dose should be ingested with a healthy and balanced diet [87]. The main food sources of vitamin C are: vegetables, especially sweet peppers (red and green), broccoli, tomatoes, potatoes, brussels sprouts, cauliflowers, cabbage and fresh and cooked leafy greens [88]. Once ingested, vitamin C meets redox reactions because it is a good electron donor. It is a reversible redox system, that leads to the reconversion of its oxidized form (dehydroascorbic acid) to its reduced form (ascorbic acid) by means of the enzymatic action of glutathione reductase and nicotinamide adenine dinucleotide (NADH) oxidase. It holds an antioxidant role in different systems and apparatus [85,89]. The bioavailability of vitamin C is mediated by several mechanisms and factors, including the efficiency of the intestinal absorption and digestive processes [88]. Vitamin C performs several functions: it maintains physiological blood vessel activity and oral and gingival health, and it favors animal and vegetable iron absorption, it helps in the growth and the repair of bone and connective tissues, and it intervenes in wound healing processes [85].

In detail, vitamin C plays a pivotal role in some pathways, such as synthetic, metabolic, hormonal, and structural mechanisms. Indeed, it intervenes in: (i) biosynthesis of collagen, because it catalyzes the conversion of proline and lysine, which are both basal components of collagen [90]; (ii) synthesis of neurotransmitters, such as norepinephrine and serotonin [91]; (iii) hormonal synthesis of vasopressin, oxytocin, adrenocorticotropic hormone (ACTH), thyrotropin-releasing hormone (TRH), steroid hormones of the adrenal cortex [92,93]; (iv) biosynthesis of bile acids, proteins, enzyme complexes involved in cellular respiration and control of lipid metabolism [94,95,96]; (v) control of intestinal absorption, transport and utilization of iron [97]; (vi) control of histamine release during allergic response [98]; (vii) cellular antioxidant activity against reactive oxygen species (ROS)-induced and UV-induced damage [99]; (viii) immune function [100].

Among patients with CKD, those in ESRD are more likely to develop vitamin C deficiency due to dietary restrictions, characterized by a lower intake of high-potassium content foods, including citruses, and due to its loss through dialysate and to gastrointestinal symptoms, such as nausea and anorexia [20,101,102].

The low molecular mass of ascorbic acid facilitates its rapid diffusion through the dialysis membrane and its elimination within the dialysate [103]. Papastephanidis et al. demonstrated that HD patients have an average concentration of ascorbic acid four times lower in serum, collected before dialysis treatment, compared to healthy subjects [103].

In fact, during the dialysis treatment both water-soluble toxins and non-toxic micronutrients, such as ascorbic acid, are eliminated from the bloodstream [103]. This lack in renal replacement therapy-RRT patients (namely in HD and in peritoneal dialysis) could also be related to a lower blood concentration of prealbumin and to an increased inflammatory state, which could be monitored by the concentration of highly sensitive C-reactive protein (hs-CRP) [104]. Another factor that contributes to lower levels of vitamin C is the OS observed in uraemia [105]. In this regard, mitochondrial dysfunctions play a pivotal role in inducing OS in CKD patients and they are characterized by reduced enzyme activity of cytochrome C oxidase and increased expression of NADH [106]. Therefore, the mitochondrial transmembrane potential is altered, resulting in increased ROS production at the level of the respiratory chain [107] with damage to both the mitochondria and the surrounding cellular structures [108]. Even if OS can occur in both HD and PD patients, the excess of ROS is more frequent in HD patients, due to membrane-blood contact and blood-dialysate contact [102]. This phenomenon is also related to the type of dialysis membrane used during the dialytic treatment; in fact, each type of membrane is characterized by a different biocompatibility [109,110,111]. Dialysis membranes are divided into those with lower biocompatibility (such as cuprofan) and those with higher biocompatibility (like polysulfone) [109,110,111]. The former can cause greater stimulation of the immune system than the latter, amplifying the chronic inflammatory state of CKD [109,110,111].

Therefore, the LGI state and the OS linked to CKD are certainly involved in the etiopathogenesis of vitamin C deficiency [104]. Patients with a lack of vitamin C are more likely to suffer from gingival bleeding and seem to be more predisposed to PD [112,113]. The latter is linked to the vitamin C levels, as it has a fundamental role in the collagen synthesis [114], because it is necessary for the hydroxylation of proline and lysine, whose hydroxy forms are the essential precursors of the collagen protein. In addition, vitamin C is involved in maintaining the integrity of connective tissue, encouraging the wound healing process and the formation of scar tissue. Therefore, its deficiency seems to cause gingival bleeding. All the oral and systemic complications, that are attributable to this vitamin deficiency, can be brought together into the clinical picture of scurvy. Scurvy is a disease caused by vitamin C deficiency [115] and is linked to the clinical features of avitaminosis. This pathological condition is characterized by gingival hemorrhagic-ulcerative manifestations, cachexia, skin hemorrhage and extensive hemorrhagic manifestations. Vitamin C plasma levels can decrease, not only due to food deficits, but also from infections and post-surgical stress [116], because, as described previously, this vitamin is involved in tissue repair processes [117,118,119]. The diagnosis of scurvy is based on the typical clinical signs of hypovitaminosis and on the resolution of these signs after the intake of vitamin C. Vitamin C serum levels lower than 0.2 mg/dL indicate scurvy [120] and its treatment relies on an appropriate intake from diet and/or correct vitamin C supplementation [120].

In addition to scurvy, it is possible to observe the onset of PD secondary to vitamin C deficiencies. In fact, the anti-inflammatory, antioxidant and regenerative functions of vitamin C suggest the main role of the latter in counteracting the inflammatory response in PD [90,99,100]. Indeed, periodontal pathogens not only induce inflammation and local tissue injury, but are also related to systemic inflammatory state and the OS [121]. Therefore, vitamin C can be supplemented in order to decrease the severity of PD and to improve its prognosis. Vitamin C counteracts the production of ROS and improves periodontal regeneration, promoting periodontal ligament progenitor cell differentiation [122,123]. In this regard, a longitudinal study related grapefruit juice consumption, rich in vitamin C, with decrease in gingival bleeding [124]. Moreover, another study demonstrated that decreased formation of deep periodontal pockets was correlated to higher consumption of citruses [125]. Furthermore, a vitamin C local injection can help to reduce local inflammation, improving the healing process, increasing collagen production and enhancing gingival circulation [126]. Vitamin C oral supplementation, combined with non-surgical treatment of chronic PD, should prove to be effective in countering periodontal inflammation [127].

### 5.2. The Role of B-Group Vitamins

B-group vitamins include thiamine (vitamin B1), riboflavin (vitamin B2), niacin (vitamin B3), pantothenic acid (vitamin B5), pyridoxine (vitamin B6), biotin (vitamin B7 or B8), folic acid (vitamin B9) and cobalamin (vitamin B12). Vitamin B complex is necessary for cell growth and for several metabolic processes [128]. Each component of B-group vitamins is characterized by a specific chemical structure and performs different functions. In particular, B1, B2, B3 vitamins and biotin are involved in several metabolic mechanisms of energy production, vitamin B6 is essential in the metabolism of amino acids and vitamin B12 and folic acid intervene in the cell cycle. Among B-group vitamins, relevant functions are carried out by folic acid and vitamin B12. The first is derived from polyglutamates that are, in turn, split into mono-glutamates in the bowel and afterwards transported thought mucosa by a specific carrier [129], while the second is assumed with foods, such as cobalamin. The latter binds the intrinsic factor, that is produced by gastric cells, and this complex reaches the ileum where it is internalized by enterocytes. Vitamin B12 is filtered by the glomerulus and is excreted in minimal part with urine [129,130]. The deficiency of B-group vitamins is associated with increased adverse outcomes [131]. In particular, hypovitaminosis is related to an enhancement of colon and breast cancer, megaloblastic anemia, and coronary heart disease [131,132]. CKD patients are at high risk of developing a lack of these vitamins due to polypharmacy, dietary restrictions, uremic gastropathy, and their removal during HD treatment [133]. In fact, among the numerous factors that induce B-group vitamins deficiency in the nephropathic patient, there is the assumption of specific drugs that reduce vitamin absorption, such as the prolonged use of metformin and of proton pump inhibitors. Moreover, the dietary-nutritional restrictions of the CKD patients, together with the CKD-related comorbidities (such as the slowing of gastric emptying secondary to autonomic neuropathy, uremic gastritis, alteration of intestinal permeability and of gut microbiota) predispose these patients to a greater risk of B-group vitamins deficiency compared to the general population [134,135,136]. The HD treatment itself represents a further risk factor for lack of B-group vitamins. In particular, this phenomenon is more evident for folic acid, the low molecular size of which allows it to be removed and cleared during dialysis treatment [21]. On the contrary, the removal of vitamin B12 is more difficult during the dialysis session, as it has a greater molecular size [21].

In CKD, homocysteine (Hcy) values rise when renal function declines towards terminal uremia, even if the causes are still unclear [137,138,139,140]. Up to now, it is well-known that the enzymes responsible for the trans-sulphuration of Hcy into cysteine, such as cystathionine-β-synthase and cystathionine-γ-lyase, are mainly expressed in the kidney and the progression of renal dysfunction results in a decline of this metabolic pathway. Furthermore, among the cofactors involved in the metabolism of Hcy, folic acid, vitamin B12 and vitamin B6 play key roles. Folic acid acts as a donor of methyl groups for methylenetetrahydrofolate reductase, an important enzyme involved in the Hcy re-methylation process, while vitamin B6 is a cofactor of the previously mentioned enzymes (cystathionine-β-synthase and cystathionine-γ-lyase). Finally, vitamin B12 is a cofactor for the methionine synthase enzyme that is involved in the re-methylation methionine pathway [141]. Therefore, a deficiency of B-group vitamins represents an important risk factor for an increase in Hcy levels [137].

Another important action of folic acid is the amelioration of endothelial function in an independent manner by Hcy levels. In fact, folic acid is involved in the reduction of OS, in the increase of endothelial nitric oxide (NO) synthase activity, in the enhancement of NO levels and in the improvement of myocardial function [141]. Vitamin B12 deficiency can also be responsible for megaloblastic anemia and neuropsychiatric diseases [142]. Furthermore, the use of B-group vitamin supplements seem to be an effective nutraceutical in restoring periodontal health [143,144].

The more promising results regarding the role of these vitamins in dentistry were observed in a recent randomized placebo-controlled clinical trial, in which the effectiveness of a systemic folic acid intake with scaling and root planing (SRP) in periodontitis treatment was tested. Sixty periodontitis subjects were randomly assigned into study groups and treated with SRP + folic acid (case group) and SRP + placebo (control group). Periodontal clinical parameters (plaque index-PI, gingival index-GI, PPD, CAL, gingival recession-GR) and biochemical parameters (CRP, Hcy) were analyzed in the study at baseline and in different post-treatment times. Although both groups showed an improvement in these parameters, systemic folic acid intake may be recommended as adjuvant therapy of periodontitis [145]. Among all B-group vitamins, folic acid seems to perform the most important effects on oral health. In a study conducted on the elderly by the National Health and Nutrition Examination Survey (NHANES), it was highlighted that folic acid serum levels were independently associated with PD [146]. In another epidemiologic study, conducted on Japanese non-smoking adult population, Esaki et al. analyzed dietary folate levels, instead of folic acid in serum, through a questionnaire survey. The results showed that low folate intake was associated with gingival bleeding, suggesting the importance of correct folic acid dietary intake [147]. According to “Società Italiana di Nutrizione Umana” (SINU) guidelines, the recommended daily intake of folic acid for the adult population is 320 μg/die [148]. A prospective cohort study established that low serum vitamin B12 levels are associated with worsening of periodontium quality, resulting in an increased rate of tooth decay and loss. The data reported in this study demonstrated how nutritional status can influence periodontal health [149].

For this purpose, some studies showed that eating habits can play a fundamental role in tissue healing after periodontal surgery procedures. These studies proved that B-group vitamins, vitamin D, and polyunsaturated fatty acids, such as docosahexaenoic acid (DHA) and eicosapentaenoic acid (EPA), can improve patient’s recovery after periodontal surgical therapy [150,151,152,153,154,155]. Individuals on strict vegetarian diets should consume vitamin B12 fortified foods, rather than attempting to absorb it exclusively from food sources [156,157]. A possible strategy to overcome this vitamin deficiency could be the use of cobalamin fortified toothpastes alone or in combination with vitamin B12 fortified foods.

Currently, there is a limited number of studies that investigate the role of B-group vitamins on periodontal health. Therefore, it should be interesting to perform further studies to assess the usefulness of B-group vitamins supplementation on oral health [158].

### 5.3. The Role of Vitamin D

Vitamin D belongs to the group of fat-soluble vitamins and it has a specific cytosolic receptor that allows its absorption. The human body acquires most of its vitamin D through exposure to sunlight and only 20% through diet and food supplements [159]. Plant-based foods provide vitamin D2 (ergocolecalciferol), while foods of animal origin provide vitamin D3 (cholecalciferol). Subsequently, both forms are transported to the liver, through the vitamin D binding protein, where they undergo hydroxylation in position 25, giving rise to 25 (OH) D. This precursor, for its conversion into the active form, needs to be further hydroxylated in position 1-α, giving rise to 1,25 (OH)_2_ D, from 1-α-hydroxylase, an enzyme mainly expressed in the kidney [160]. G. Kaur et al. demonstrated that a diet rich in vitamin D (serum levels 25 (OH) D > 50 nmol/L), in addition to the administration of vitamin B complex (50 mg of thiamine HCl, riboflavin, niacinamide, D-calcium pantothenate, pyridoxine HCl; 50 μg of D-biotin cyanocobalamin; and 400 μg of folate) before periodontal surgery (open flap debridement), improved the healing process of the site. This amelioration was detected by higher CAL and lower PPD, compared to patients with lower serum levels of 25 (OH) D [161]. Overall, these studies highlight the importance of dietary supplements before and after periodontal procedures for the improvement of post-operative outcomes [154,161]. The relation between vitamin D intake and oral health was extensively analyzed [88,161,162,163]. In the European Federation of Periodontology (EFP)/European Organisation for Caries Research (ORCA) workshop, it was argued that vitamin D deficiency can affect periodontal health [164]. This is supported by numerous clinical studies that showed a direct relation between vitamin D levels and periodontal health status [165,166,167,168]. A recent study assumed that subjects with chronic periodontitis have significantly lower serum levels of 25 (OH) D compared to healthy subjects, while there was no difference between the two groups in salivary levels of 25 (OH) D. The authors concluded that chronic periodontitis is strongly associated with low serum levels of 25 (OH) D [169]. On the contrary, other systematic reviews reporting data that related to a possible link between serum levels of 25 (OH) D and chronic periodontitis were conflicting [167,170], or too insufficient and controversial to reach a clear conclusion [168]. Longitudinal studies and randomized clinical trials conducted on a large population are essential to further investigate the association between vitamin D levels and the presence of PD, also to consider the possible use of vitamin D oral supplements.

In a cross-sectional study by Isola et al., it was highlighted how low levels of 25 (OH) D were significantly related to PD and how periodontal parameters (CAL, PPD, bleeding index on probing-BOP) were significantly dependent on 25 (OH) D levels (*p* < 0.001) [171].

It is important to consider the data analysis done by the NHANES III, conducted on subjects aged 50 or over, where the levels of 25 (OH) D showed a significant and inverse association with periodontal attachment loss, regardless of the value of bone mineral density [172]. Moreover, several studies have analyzed the possible relation between estimated-GFR (e-GFR) and the levels of 25 (OH) D, showing an inverse relationship among these two parameters [173,174,175]. For this purpose, an interesting case-control study demonstrated that CKD patients under conservative therapy with lower median values of 25 (OH) D were more predisposed to PD onset [18]. CKD patients present a high risk of developing vitamin D deficiency, due to both reduced sun exposure and mechanisms intrinsic to the disease itself. In particular, the hyper-phosphoremia, characteristic of HD patients, stimulates FGF-23 activity, a growth factor with hyper-phosphaturic action, which, at the same time, it, inhibits renal 1-α-hydroxylase expression and induces 24-hydroxylase expression. The latter is responsible for the transformation of vitamin D into its inactive form, namely 24,25 (OH)_2_ D [176]. Vitamin D plays a pivotal role in the regulation of bone mineral metabolism, in calcium/phosphate homeostasis, in bone remodeling and in the modulation of cell proliferation and differentiation [177]. Moreover, it exerts anti-inflammatory effects through the inhibition of inflammatory mediators [178] and it has anti-microbial effects, as 25 (OH) D is able to induce antimicrobial peptide (AMP) activity [179]. Vitamin D anti-inflammatory effects derive from the reduction of inflammatory mediator expression, such as interleukin (IL)-6 and TNF-α, IL-35, IL-17A and transforming growth factor, through nuclear factor kappa B (NF-kB) inhibition and mitogen-activated protein kinase phosphatase 1 (MKP-1) up-regulation. Pro-inflammatory cytokine reduction leads to a decrease in connective tissue destruction and to a reduction in bone resorption [166]. According to a Li et al. study, conducted in vitro on oral keratinocytes (OKF67TERT-2) incubated with lipopolysaccharide (LPS) of periodontal pathogens, 1,25 (OH)_2_ D appears to inhibit IL-6 overexpression, induced by LPS in oral epithelial cells, to increase vitamin D receptor (VDR) expression, aryl hydrocarbon receptor (AhR) activation and to repress NF-kB phosphorylation. The results show that 1,25 (OH) D could modulate the inflammatory response in periodontitis through AhR/NF-kB signaling regulation [180]. LPS, deriving from periodontal pathogenic bacteria, exerting relevant action in PD, as it induces the production of inflammatory cytokines, namely IL-6 and IL-8, TNF-α in oral epithelial cells [181]. Among inflammatory cytokines, IL-6 carries out relevant action in the reabsorption of alveolar bone. Therefore, the regulation of the inflammatory response of oral epithelial cells is considered a potential strategy for periodontitis treatment [180].

The antimicrobial effect of vitamin D results from the bond between 1,25 (OH)_2_ D and the complex VDR/vitamin D response element (VDRE), which leads to the production of AMP and β-defensins (β-Def-2 e β-Def-3) by macrophages, monocytes, cells of the human gingival epithelium (HGE) and cells of the human periodontal ligament (HPL) [166]. These antimicrobial peptides have an activity on bacterial species that cause PD, such as *Porphyromonas gingivalis*, *Fusobacterium nucleatum* e *Aggregatibacter actino-mycetemcomitans*, and they are also capable of defeating Gram-negative bacteria [182]. In addition, according to a study conducted by Hu et al., the activated form of 1,25 (OH)_2_ D decreases the amount of *Porphyromonas gingivalis* bacteria internalized by epithelial cells and monocytes [179]. CKD is related to the onset and progression of PD and this relation is probably caused by the alteration in the immune response induced by the disease itself and, currently, there is no more evidence that explains this hypothesis [2,183]. Considering the deficiency of 1,25 (OH)_2_ D induced by the decreased activity of 1-α-hydroxylase and pre-renal hypovitaminosis D [184], it would be interesting to investigate the relation between pre-renal hypovitaminosis D and PD.

Even if other studies on biological mechanisms that link vitamin D to periodontal health are necessary, it is currently assumed that the link between these two conditions relies on different factors. In the first place, low salivary levels of 25 (OH) D are associated with higher salivary levels of pro-inflammatory cytokines (transforming growth factor (TGF) β, IL-35, IL-17A, matrix metalloproteinase (MMP)-9) in PD patients, compared to healthy subjects [185]. Secondly, low levels of vitamin D could interfere with the antimicrobial defenses [178]. Indeed, low serum levels of 25 (OH) D are connected to the expression of human β-defensin 2 (HBD-2) and of cathelicidin reduction (in particular of its fragment LL37) in the periodontal tissue of patients with chronic PD [186].

Vitamin D deficiency can also inhibit periodontal tissue healing [187]. Furthermore, the deficiency of this micronutrient, due to an unbalanced diet, can contribute to oral dysbiosis [188]. Therefore, considering the importance of vitamin D in oral health, many studies have been performed to evaluate vitamin D supplementation efficacy and safety during or after periodontitis therapy. According to several authors, vitamin D supplementation reduces systemic inflammation by decreasing pro-inflammatory salivary cytokines and by promoting antimicrobial functions [189].

A dose-dependent effect of vitamin D supplements on periodontal parameters was found in a study conducted by Cagetti, Camoni et al. The integration of 2000 IU of vitamin D resulted in greater improvement of the gingival parameters rather than that observed in subjects treated at lower doses (1000 IU and 500 IU) [190]. On the contrary, according to a recent randomized clinical trial, the effects of vitamin D supplementation on periodontal tissues are weak and have limited clinical relevance. For this reason, further studies that evaluate the long-term efficacy and safety of vitamin D on oral clinical manifestations in chronic non-communicable disease patients are necessary [191].

## 6. The Role of Metabolic Acidosis on Onset of Oral Diseases

Among the complications of CKD, metabolic acidosis impacts on the increased onset of oral cavity diseases. In fact, this condition induces bone demineralization and increased muscle catabolism [192]. Numerous studies have confirmed that metabolic acidosis holds a pivotal role in bone demineralization and in increased protein catabolism, which are both phenomena observed in CKD [39,193,194,195]. Furthermore, it was shown that the prevalence of metabolic acidosis is inversely related to e-GFR. The reduction of the latter represents one of its most relevant risk factors, since CKD causes reduced ability to eliminate hydrogen ions [196]. In CKD patients, there are several factors that can positively or negatively influence the onset of metabolic acidosis; among these, we find [192]: (i) decrease of GFR, which doubles the risk of metabolic acidosis in CKD stage III and increases the risk seven-fold in CKD stage IV; (ii) smoking increases the risk by 43%; (iii) normochromic and normocytic anemia increases the risk by 40%; (iv) the use of diuretics reduces the risk by 30%; (v) the use of ACE inhibitors increases the risk by 30%. Scientific studies showed that metabolic acidosis stimulates the proteolysis of skeletal muscle through a process called “ubiquitination”, a glucocorticoid-dependent mechanism. On the other hand, in bone tissue it induces direct structural alteration by stimulating bone resorption, mediated by osteoclasts, and by inhibiting bone formation, mediated by osteoblasts. Moreover, it alters both the biological functions and the serum concentration of PTH and vitamin D. These conditions negatively impact on the health of oral cavity and they become a risk factor for the development of periodontitis, because they modify the stability of periodontal support tissues [39,194,197,198]. In particular, bone tissue in the presence of metabolic acidosis tries to restore the physiological acid-base balance by buffering the excess of hydrogen ions, which are increased in CKD patients. The mechanism through which the bone tissue performs this function is the bond of hydrogen ions to its surface, which is equipped with negative electrical charges [197].

Studies conducted on bone autopsy fragments, or bone findings removed surgically from CKD patients or from subjects with normal renal function (control group), showed reduction of calcium carbonate content of the bone tissue in nephropathic patients, compared to the control group. Furthermore, this reduction was related to the duration of uremia and it is probably induced by the damage caused by metabolic acidosis on bone tissue [199].

## 7. The Relation between Low-Grade Chronic Inflammation and Oral Diseases

Periodontitis is associated with different comorbidities, which recognize a common pathogenetic denominator in LGI. LGI is characterized by the chronic systemic production of inflammatory factors and it is one of the risk factors for various chronic diseases, such as diabetes mellitus, CV diseases, CKD, and cancer [200,201]. Compared to healthy subjects, CKD patients may have a persistent low-grade inflammatory status, which, combined with premature aging, called “inflammageing”, is a distinctive sign of the uremic phenotype and it contributes to an altered state of health, to reduced quality of life and to an increased risk of all-cause mortality [202]. In patients affected by PD, a “two-way relationship” may occur between oral manifestations and chronic degenerative non-communicable diseases. In the presence of these pathological conditions, the tissues and the cells respond with local production of pro-inflammatory mediators (IL-1, IL-6, TNF-α, prostaglandin E2); these can spread into the systemic level, inducing and amplifying the systemic inflammatory state. At the same time, the systemic inflammation can influence periodontal health [203]. Several clinical studies evaluated the association between systemic levels of inflammatory markers and the risk of PD [204,205,206]. In a study conducted by Cecoro et al., the authors evaluated the possible link between periodontitis, LGI and general health. The obtained data showed that LGI and the onset of systemic inflammatory phenotype could represent the common substrate of many chronic inflammatory diseases, such as periodontitis, obesity, diabetes mellitus, CV diseases, metabolic syndrome, CKD, etc. According to the conclusions of this study, the bidirectional relation between periodontitis and general health is confirmed by LGI involvement [200]. Therefore, by understanding the basic mechanisms of these complex reciprocal interrelationships and considering that PD could contribute to the development of LGI phenotype, it is essential to include periodontal patients in supportive periodontal therapy. In a periodontal patient with CKD, the “two-way relationship” appears to be established. Dysbiosis of the oral microbiota could cause an initial local inflammation which could then extend systemically; in addition, it could exacerbate the chronic inflammation already present in CKD, thus accelerating the progression of CKD itself. Furthermore, CKD could cause an increase of salivary pH, tartar deposits, xerostomia and alterations in immune response [207]. To these mechanisms, other factors, such as polypharmacotherapy, the severity of CKD, renal replacement therapy, and poor oral hygiene could be added. Further studies are needed to assess the underlying mechanisms of the bidirectional link between periodontitis and CKD.

## 8. Conclusions

The nutritional status of CKD patients seems to be altered by several risk factors, such as PEW, uremic gastropathy, polypharmacy, dietary restrictions, renal replacement therapy, metabolic acidosis, lack of vitamins, and LGI status, which can also contribute to the development of PD. Furthermore, CKD and periodontal patients should be managed by a multidisciplinary professional team, who could evaluate the possible copresence of these two pathological conditions, that negatively influence each other, and set up therapeutic strategies to treat them. Once these patients have been identified, they should be included in a follow-up program that provides periodic checks in order to treat these pathological conditions. It would be advisable to plan an interdisciplinary collaboration path between dialysis centers and dental facilities and to raise awareness of the need for preventive dental care in dialysis patients. In fact, the treatment of PD could reduce the systemic inflammatory burden induced by the oral pathology, the risk of atherosclerotic vascular disease and the mortality of HD patients.

## Figures and Tables

**Figure 1 nutrients-14-02002-f001:**
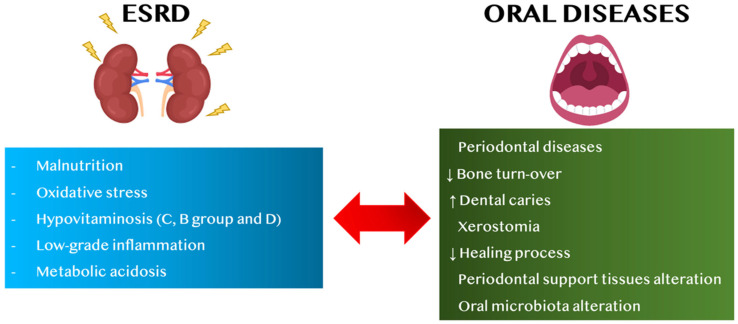
Link between ESRD and oral diseases, in adults. Abbreviations: ESRD, end-stage renal disease, ↑ increase, ↓ decrease.

**Figure 2 nutrients-14-02002-f002:**
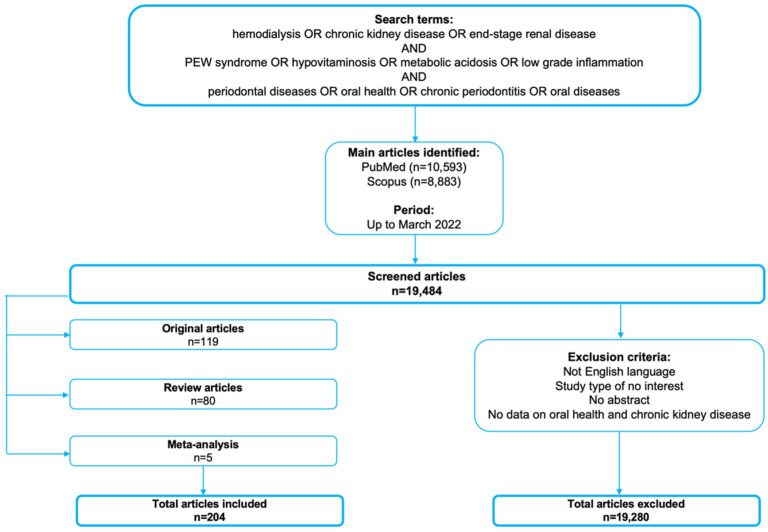
Article search methodology.

**Figure 3 nutrients-14-02002-f003:**
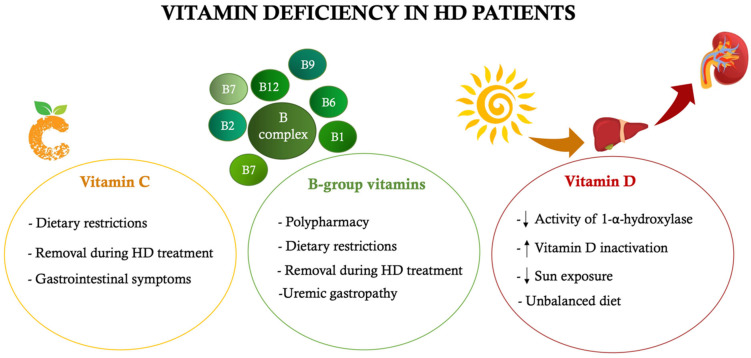
Main causes of vitamin deficiency in hemodialysis patients. ↑increase, ↓ decrease. HD: hemodialysis.

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
