# Peer review of "The Impact of Chronic Kidney Disease on Nutritional Status and Its Possible Relation with Oral Diseases"

_nutrients, 2022, doi:10.3390/nu14102002_

Round 1
Reviewer 1 Report
In their review, entitled “The impact of haemodialysis on nutritional status and its possible correlation with oral diseases”, the Authors presented the problem of the interrelationship between end-stage renal failure and the severity of periodontal diseases mediated mainly by nutritional deficiencies and the severity of inflammation.
The work is written understandably and the described mechanisms and dependencies are fully explained. However several elements need to be developed, especially in terms of the clinical and social significance of the problem described and the conclusions drawn.
Detailed notes are described below:
- Title
I suggest changing the words in the title of the work. First, the described disorders leading to periodontitis are caused by many factors resulting from the underlying disease (ESKD), not only from the hemodialysis treatment used. The title should take this fact into account.
Second, the use of the term "correlation" does not seem very accurate. In scientific publications, this term is most often used in the context of statistical analysis. Therefore, it is more justified in this context to use the word 'Relation'.
I suggest that this term be changed consistently throughout the manuscript.
- line 45
Term “bimodal correlation” should be replaced by a bi-directional dependency/mutual relationship. Bimodal indicates two modes and is correct for describing the histograms.
- Lines 61-64 and Figure 1.
Figure 1 is a good idea to summarize the Introduction however, its structure and the idea conveyed are not consistent with the description in the paragraph above. Figure 1 shows a unidirectional relationship between ESDR and oral diseases, while in the manuscript the authors indicate that the relationship is bidirectional. It is necessary to edit the Figure and insert arrows in both directions indicating what factors occur in a given relationship.
- Introduction
The Introduction chapter consists mainly of the results of several works pointing to the main theses made in the manuscript. It is advisable to develop this chapter in the following aspects:
- the clinical and social significance of periodontal diseases for people suffering from CKD
- a short description of the ESHD on which the rest of the work is mainly focused
- an indication of the goal which guided the review of the literature and which the authors wanted to achieve
- presenting the work plan and the sequence of aspects discussed so that the reader is more oriented in the content of the work and encouraged to read the work further
- Search methods
The method of searching and qualifying works for the preparation of a review is significantly insufficient. There is no information on how many publications were found, which ones were rejected, and on what basis, what types of publications were included. It is advisable to prepare this chapter in accordance with, for example, the PRISMA Flow Diagram.
http://www.prisma-statement.org/PRISMAStatement/FlowDiagram
- line 124
It is difficult to agree with the authors' opinion that “metabolic acidosis” was discussed earlier. The phrase "as previously described" is not valid.
- lines 142-145
This sentence needs to be rewritten as it is imprecise. Increasing the TNF concentration and the consequent increase in CRP do not lead to the development of inflammation.
- Figure 2
If the loss of vitamin C and B vitamins during dialysis occurs through the same mechanism, the description of this phenomenon should be the same in the figure.
- line 296
In this context, it is more appropriate to use the term “the concentration” instead of “the dosage”.
- line 305
In a new paragraph, it is necessary to precisely indicate what phenomenon it describes. My guess is "it" means Oxidative Stress
- lines 408-412
Such a detailed description of the research methodology is unnecessary. It is enough to provide the most important conclusions, readers interested in the study can find them thanks to References.
- Conclusions
lines 636-641
This conclusion is not apparent from the content of the thesis. As I mentioned earlier, the work lacks an introduction to the described issues. What potential benefits will patients identified and included in potential follow-up programs achieve?
General comment:
Some of the abbreviations are not expanded on first use:
- line 72: PEW
- line 305: PD
Author Response
Rome, 2nd May 2022
Dear Editor,
all the corrections have been written in red color in the revised version of the manuscript. We reviewed the entire manuscript according to the reviewer’s comments.
We would like to thank reviewer #1 for his/her comments.
He/she wrote:
- “I suggest changing the words in the title of the work. First, the described disorders leading to periodontitis are caused by many factors resulting from the underlying disease (ESKD), not only from the hemodialysis treatment used. The title should take this fact into account.”
We changed the title in “The impact of chronic kidney disease on nutritional status and its possible relation with oral diseases”, as suggested.
- “Second, the use of the term "correlation" does not seem very accurate. In scientific publications, this term is most often used in the context of statistical analysis. Therefore, it is more justified in this context to use the word 'Relation'. I suggest that this term be changed consistently throughout the manuscript.”
We replaced “correlation” with “relation”, as suggested.
- “line 45, Term “bimodal correlation” should be replaced by a bi-directional dependency/mutual relationship. Bimodal indicates two modes and is correct for describing the histograms.”
We replaced “bimodal correlation” with “bi-directional”, as suggested.
- “Lines 61-64 and Figure 1, Figure 1 is a good idea to summarize the Introduction however, its structure and the idea conveyed are not consistent with the description in the paragraph above. Figure 1 shows a unidirectional relationship between ESDR and oral diseases, while in the manuscript the authors indicate that the relationship is bidirectional. It is necessary to edit the Figure and insert arrows in both directions indicating what factors occur in a given relationship.”
We corrected the figure 1, as suggest.
- “The Introduction chapter consists mainly of the results of several works pointing to the main theses made in the manuscript. It is advisable to develop this chapter in the following aspects:
The clinical and social significance of periodontal diseases for people suffering from CKD”
We added this aspect in the introduction, as requested.
A short description of the ESRD on which the rest of the work is mainly focused.
We added the description of ESRD in the introduction, as requested.
An indication of the goal which guided the review of the literature and which the authors wanted to achieve. Presenting the work plan and the sequence of aspects discussed so that the reader is more oriented in the content of the work and encouraged to read the work further.
We added the aims and we described all the points discussed in the review.
- “The method of searching and qualifying works for the preparation of a review is significantly insufficient. There is no information on how many publications were found, which ones were rejected, and on what basis, what types of publications were included. It is advisable to prepare this chapter in accordance with, for example, the PRISMA Flow Diagram.”
We have improved the chapter of searching methods, as required.
- “line 124, It is difficult to agree with the authors' opinion that “metabolic acidosis” was discussed earlier. The phrase "as previously described" is not valid.”
We deleted the sentence “as previously described”, as suggested.
- “lines 142-145, This sentence needs to be rewritten as it is imprecise. Increasing the TNF concentration and the consequent increase in CRP do not lead to the development of inflammation.”
We rephrased and we corrected the sentences.
- “Figure 2, If the loss of vitamin C and B vitamins during dialysis occurs through the same mechanism, the description of this phenomenon should be the same in the figure.”
We have corrected the figure, as requested.
- “line 296, In this context, it is more appropriate to use the term “the concentration” instead of “the dosage”.
We replaced “the dosage” with “concentration”, as requested.
- “line 305, In a new paragraph, it is necessary to precisely indicate what phenomenon it describes. My guess is "it" means Oxidative Stress”
We corrected the sentence as suggested.
- “lines 408-412, Such a detailed description of the research methodology is unnecessary. It is enough to provide the most important conclusions, readers interested in the study can find them thanks to References.”
We deleted the unnecessary details, as requested.
- “lines 636-641, This conclusion is not apparent from the content of the thesis. As I mentioned earlier, the work lacks an introduction to the described issues. What potential benefits will patients identified and included in potential follow-up programs achieve?”
We have improved the conclusions, as requested.
- “Some of the abbreviations are not expanded on first use: line 72: PEW, line 305: PD.”
We corrected all the abbreviations in the text.
Best regards,
Prof. Annalisa Noce

Reviewer 2 Report
In this review article, “The impact of haemodialysis on nutritional status and its possible correlation with oral diseases,” the authors reviewed the related nutritional factors between CKD patients and oral diseases. The manuscript was written well. I just have one minor point for this study.
- Some sentences should be combined with a paragraph. You can see more paragraphs just presented one sentences in the manuscript.
Author Response
Rome, 2nd May 2022
Dear Editor,
all the corrections have been written in red color in the revised version of the manuscript. We reviewed the entire manuscript according to the reviewer’s comments.
We would like to thank reviewer #2 for his/her comments.
We would like to thank Reviewer 1 for his/her comments. All corrections are shown in red in the text.
- “Some sentences should be combined with a paragraph. You can see more paragraphs just presented one sentences in the manuscript.”
We merged some sentences, as suggested.
Best regards,
Prof. Annalisa Noce

Round 2
Reviewer 1 Report
The authors have made significant efforts to improve the manuscript based on the previous review.
Congratulations on your very good work on the multidisciplinary problem.